# Accelerating Part-Scale Simulation in Liquid Metal Jet Additive Manufacturing via Operator Learning

**Søren Taverniers,**[1] **Svyatoslav Korneev,**[1] **Kyle M. Pietrzyk,**[1] **Morad Behandish**[1]

[1] Palo Alto Research Center (PARC), 3333 Coyote Hill Road, Palo Alto, CA 94304, USA
moradbeh@parc.com (Morad Behandish)

## Abstract

Predicting part quality for additive manufacturing (AM) processes requires high-fidelity numerical simulation of partial differential equations (PDEs) governing process multiphysics on a scale of minimum manufacturable features. This makes part-scale predictions computationally demanding, especially when they require many small-scale simulations. We consider drop-on-demand liquid metal jetting (LMJ) as an illustrative example of such computational complexity. A model describing droplet coalescence for LMJ may include coupled incompressible fluid flow, heat transfer, and phase change equations. Numerically solving these equations becomes prohibitively expensive when simulating the build process for a full part consisting of thousands to millions of droplets. Reduced-order models (ROMs) based on neural networks (NN) or k-nearest neighbor (kNN) algorithms have been built to replace the original physics-based solver and are computationally tractable for part-level simulations. However, their quick inference capabilities often come at the expense of accuracy, robustness, and generalizability. We apply an operator learning (OL) approach to learn a mapping between initial and final states of the droplet coalescence process for enabling rapid and accurate part-scale build simulation. Preliminary results suggest that OL requires order-of-magnitude fewer data points than a kNN approach and is generalizable beyond the training set while achieving similar prediction error.

## Introduction

Droplet-scale dynamics for LMJ (Sukhotskiy et al. 2017; Bikas, Stavropoulos, and Chryssolouris 2016) can be modeled by coupled incompressible and immiscible multi-phase fluid flow, (convective and conductive) heat transfer, and solidification equations (Korneev et al. 2020), which can be spatially discretized using a finite volume (FV) approach and solved by time integration using computational fluid dynamics (CFD) platforms such as OpenFOAM (Jasak et al. 2007). Such simulations, in conjunction with experimental calibration of the material properties, can provide an accurate prediction of the droplet-scale dynamics. However, the computations can slow down due to constraints on the temporal step that guarantee stability during a numerical simulation, e.g., the Courant–Friedrichs–Lewy (CFL) condition.

Part-scale build simulation requires calling the droplet-scale solver numerous times in a sequential loop with a moving domain of interest, where the final conditions of each droplet coalescence simulation serve as initial conditions to the next one. These conditions include values for phase, velocity, pressure, and temperature. In the context of LMJ, computing the coalescence of a single droplet, with a diameter of a few hundred microns, may take an FV solver up to an hour on a 96-core cluster [1], while build simulation for 3D printed parts consisting of thousands to millions of droplets becomes prohibitively expensive, if not impractical.

Previously, (Korneev et al. 2020) constructed a ROM of the droplet-scale physics of the LMJ process based on a k-nearest neighbors (kNN) search within a set of data generated offline by a coupled multiphysics solver implemented in OpenFOAM. This algorithm can estimate the shape of solidified droplets on an arbitrary substrate at a speed of $\sim 33$ droplets per second on the same 96-core cluster, a significant improvement compared to the high-fidelity OpenFOAM solver. Applying the ROM recurrently along a sampled toolpath, (Korneev et al. 2020) estimated the shape of a part consisting of $\sim$50,000 droplets, a result that would be impractical to achieve using OpenFOAM. Although using this ROM in place of OpenFOAM yielded orders of magnitude in speed up, unfortunately, the kNN search extrapolated poorly for out-of-training data, requiring a large data set to cover all possible substrate geometries, thereby offsetting the gains from the achieved speedup.

Here we present an improved ROM to enable part-scale build simulations for LMJ using operator learning (OL) to approximate the droplet-scale physics. Rather than approximating the solution to the governing system of PDEs for a particular instance of initial/boundary conditions (ICs/BCs), as is done, for example, in physics-informed NNs (PINNs) (Raissi, Perdikaris, and Karniadakis 2019), OL allows one to learn the *operator* that maps the initial condition of a single droplet deposition in the moving subdomain to the final condition at the end of the deposition. The same trained operator can be used to predict this initial-to-final condition mapping across numerous instances of the problem with the same PDEs and BCs, but different ICs. While a similar approach was already considered by the authors of (Korneev

---

[1] Amazon AWS c5 instance, specifically c5.24xlarge.

et al. 2020) using a fully-connected feed-forward NN, the quadratic scaling of the number of network weights with the number of degrees of freedom (in this case, spatial grid size) required a prohibitively large network size for accurate predictions, making failures common after only a few sequentially deposited droplets. Instead, here we implement the recently developed Fourier neural operator (FNO) (Li et al. 2020, 2021), a deep NN which learns a kernel integral operator related to the PDE's Green's function (or a generalization thereof, for nonlinear PDEs). This approach was found to yield a much smaller test error for the same amount of training data (Li et al. 2020). Moreover, FNO uses the convolution theorem to learn this operator in the Fourier domain, enabling speedup through the use of the Fast Fourier Transform (FFT) algorithm.

Below, we briefly review the *moving subdomain* approach used in (Korneev et al. 2020) in conjunction with a droplet-scale simulator of droplet-substrate coalescence, using either FV-based CFD (in OpenFOAM) or a kNN-based ROM (in Cython) to obtain a part-scale as-manufactured shape predictor. We then show how replacing kNN with FNO enables faster part-scale simulation at comparable accuracy with significantly fewer training data points.

## Reduced-Order Modeling for LMJ

The high-fidelity LMJ model can be decomposed into a series of single-droplet coalescence events applied along the toolpath (Fig. 1). The ICs for every coalescence event consist of a hot liquid droplet of spherical shape (pictured in red) captured by a phase field, its initial velocity, and a substrate of arbitrary shape. The substrate, on average, is composed of solid material. After hitting the substrate, the droplet solidifies and coalesces with the substrate surface; previous droplets that have coalesced with the substrate become part of the ICs for the next droplet. Figure 2 shows a time sequence of the coalescence for two consecutive droplets.

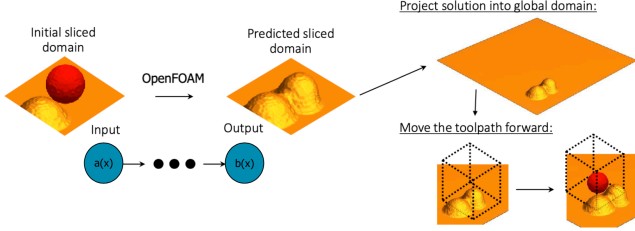

Figure 1: A moving subdomain approach for sequential deposition of droplets along a toolpath. Red indicates a liquid phase, while orange indicates a solid phase.

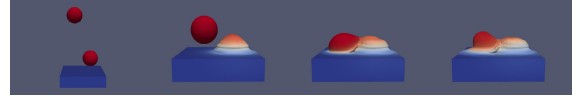

Figure 2: Sequential deposition of two initially liquid droplets onto a substrate. Red indicates hotter zones, while blue indicates cooler zones. Source: (Korneev et al. 2020).

For the LMJ process, the droplet temperature is slightly above the solidification temperature. This low temperature difference minimizes residual stresses and eliminates warping of the final geometry. The absence of warping simplifies the physics of the LMJ process to the incompressible flow and heat transfer equations (Korneev et al. 2020).

High-fidelity numerical solutions of the droplet physics can be obtained using a finite volume (FV), volume of fluids (VoF) scheme in OpenFOAM. However, these simulations can become prohibitively expensive at the part scale, where thousands or even millions of droplets need to be deposited. This prompted (Korneev et al. 2020) to construct a kNN search algorithm that could predict the droplet coalescence at a fraction of the computational cost of the Open-FOAM solver. First, a set of 9,000 samples was generated with the OpenFOAM solver, where the input and output included solid and liquid phase variables—from which the gas phase can be obtained, since, by definition, they must add up to unity—before and after the simulation, i.e., when the liquid droplet is slightly above the substrate and when it hits and merges with it after solidification, respectively (Fig. 1). When presented with a new input, the training set was searched for its kNNs and the predicted output was computed via averaging of the outputs corresponding to these neighbors (Korneev et al. 2020).

While an accelerated version of the kNN algorithm in (Korneev et al. 2020) could predict a single droplet deposition in about 0.03s (i.e., a 20,000x speedup compared to OpenFOAM) on the same 96-core cluster, this was still longer than the actual deposition time on the machine (0.01s for a 100Hz deposition frequency). Moreover, the method was not designed to generalize beyond the training set. To rectify these shortcomings, here we present an OL based approach to map initial to final conditions in the moving subdomain. We use an updated data set, obtained from Open-FOAM simulations, with an improved multiphysics model involving experimentally calibrated parameters.

## Operator Learning for LMJ

The underlying idea of OL for scientific computing is to approximate maps $\mathcal{M}^\dagger$, between infinite-dimensional function spaces, representing solution operators of initial/boundary-value problems. More concretely, we aim to construct a parametric map:

$$\mathcal{M}_\lambda : \mathcal{A} \to \mathcal{B}, \quad \lambda \in \Lambda \quad (1)$$

for a finite-dimensional parameter space $\Lambda$ by choosing an "optimal" value $\lambda^\dagger \in \Lambda$ such that $\mathcal{M}_{\lambda^\dagger}$ represents the best approximation to $\mathcal{M}^\dagger$ in some sense (e.g., minimizing a least-squares error). Here $\mathcal{A} = \mathcal{A}(\Omega; \mathbb{R}^{d_a})$ and $\mathcal{B} = \mathcal{B}(\Omega; \mathbb{R}^{d_b})$ are separable Banach spaces of functions defined on some bounded, open set $\Omega \subset \mathbb{R}^d$. For example, a function $a \in \mathcal{A}$ can be an initial condition (say at time $t = 0$) or a parameter of a PDE, and $b = \mathcal{M}^\dagger(a)$ is the solution of that PDE at some time $t > 0$ (Li et al. 2020).

While the PDE itself is typically defined locally, its solution operator has non-local effects that can be described by integral operators. This inspired the authors of (Li et al. 2020) to approximate the (possibly generalized) Green's

function of a problem's governing PDE by a graph kernel network. In (Li et al. 2021), the same authors then interpreted this kernel as a convolution operator through the architecture visualized in Fig. 3 and briefly reviewed in the Appendix. This approach enables a finite-dimensional parametrization of the input/output functions via a truncated Fourier basis.

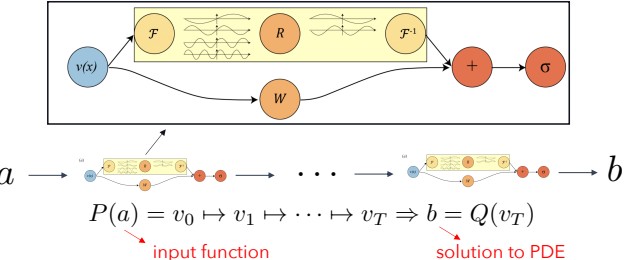

Figure 3: Fourier neural operator (FNO) architecture. Adapted from (Li et al. 2021).

Identifying $a(\mathbf{x}) \in \mathbb{R}$ and $b(\mathbf{x}) \in \mathbb{R}$ for $\mathbf{x} \in \Omega \subset \mathbb{R}^3$, where $\Omega$ is the moving subdomain, as the initial and final conditions, respectively, specified through the combined solid, liquid, and brass[2] phase fractions at $t = 0$ and $t = 0.0025$s for a 400 Hz deposition frequency, we replace kNN with FNO to sequentially deposit droplets along the toolpath as before.

We train the FNO surrogate using 770 input/output pairs generated by simulations of 4 pyramid parts (620 data points) and 1 hollow cylinder part (150 data points), where the latter is deemed useful by numerical experimentation to handle part geometries with thin features. We test the resulting model using 324 input/output pairs generated by simulations of a cube part (i.e., different from the training set). We repeat this process for different sets of hyperparameters—namely, Fourier layer width and number of retained Fourier modes—until a satisfactory combination is produced.

Training of and inference with the FNO surrogate was done using PyTorch code made available on the public domain under the MIT License (Li, Cao, and Griffiths 2021) by (Li et al. 2021). To take advantage of GPU-accelerated FFT, training and prediction were done on an NVIDIA RTX 3090 GPU.

## Results

Figure 4a shows the distribution of errors on the cubes test data set for an optimized set of hyperparameters—namely, Fourier layer width and number of retained Fourier modes. The distribution of errors is skewed toward smaller values than the average of 16.7% with a mode slightly above 10%.

Following this test set validation, we use the trained FNO model in conjunction with the moving subdomain method for inference of single lines of droplets sequentially deposited with spacings of a few hundred microns. Counter-

<hr/>

[2]We assume the substrate to be made of brass, to resemble the build plate of the LMJ 3D printer, while the droplets are made of aluminum.

parts computed by the CFD solver in OpenFOAM serve as the "ground truth." Figure 4b visualizes the FNO prediction and corresponding OpenFOAM result for droplet spacings $S_{\text{norm}}$ equal to 62.72% (1), 89.61% (2) and 116.49% (3) of the droplet diameter $D$. For each of these cases, the left isosurface is predicted by FNO and colored according to the distance (normalized with respect to $D$) between each vertex on this surface and the vertex on the OpenFOAM isosurface (right, in gray) closest to that point. The largest of these distances corresponds to the so-called Hausdorff distance $d_{\text{H}}$, which is visualized in the left part of Fig. 4b for all considered droplet spacings as $d_{\text{H,norm}} = d_{\text{H}}/D$ (in %). Although $d_{\text{H,norm}}$ can reach values up to 30%, from the distance heat maps on the right we can see that the majority of the relative errors is less than 15%.

LMJ-generated parts are printed by layering many droplet lines such as those visualized in Fig. 4b on top of each other. Hence, the first step in assessing FNO's ability to predict such parts is to focus on only a few layers of stacked droplet lines, as shown in Fig. 5 for a normalized droplet spacing $S_{\text{norm}}$ of 89.61%. In dark gray, we show the prediction of FNO trained on the mixed training set consisting of both pyramid and hollow cylinder parts detailed in the previous section. Compared to the prediction (in blue) of FNO trained on 1,460 data points from only pyramid parts, we note a clear qualitative improvement in the prediction accuracy. This could be explained by the fact that inclusion of the hollow cylinder data in the training set improves FNO's learning of thin-wall scenarios, and allows it to outperform its counterpart trained on a larger, but less diversified, set of pure pyramid data.

Figure 6 shows FNO's inference of a gear-shaped part generated by 16,000 droplets with $S_{\text{norm}} = 89.61\%$. A more detailed view of the upper section reveals that FNO is capable of predicting repeated layers of droplet lines, including those along part edges, although some imperfections can be seen along both the inner and outer walls. Prediction of such a gear shape using kNN accelerated via height maps required 36,000 input-output data pairs (Korneev et al. 2020) compared to the 770 training data pairs needed for FNO, a difference of almost two orders of magnitude. Moreover, inference of a single droplet deposition took 0.03s with kNN, while FNO performs this task in ~3ms, which is one order of magnitude smaller.

## Conclusions

We implemented a surrogate model for liquid metal jetting (LMJ) based on deep learning of solution operators of the partial differential equations (PDEs) governing the droplet deposition process. Specifically, we employed the recently developed Fourier neural operator (FNO) based on approximating a kernel integral operator by a neural network (NN), and utilizing the convolution theorem to parametrize this NN in Fourier space and take advantage of Fast Fourier Transform (FFT), implemented on a GPU. We found that the FNO surrogate, trained on high-fidelity simulation data generated with multiphysics computational fluid dynamics (CFD), is capable of predicting the geometric features for single and

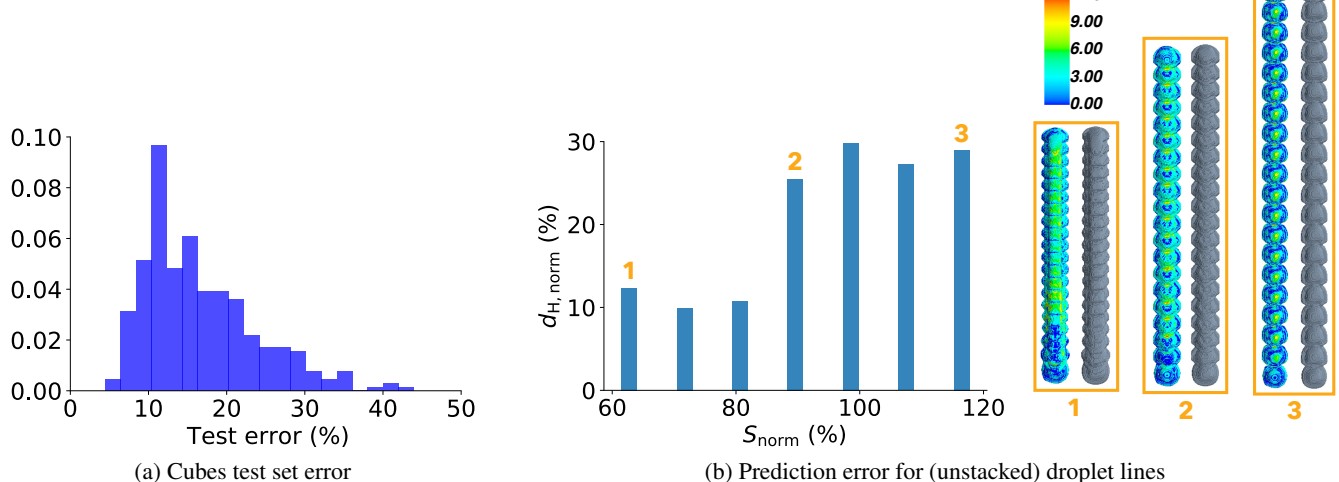

(a) Cubes test set error        (b) Prediction error for (unstacked) droplet lines

Figure 4: On the left (a), we show the error distribution for our trained FNO model on the cubes test set. On the right (b), we show the normalized Hausdorff distance $d_{\mathrm{H,norm}}$ for droplet lines of various spacings both bigger and smaller than the droplet diameter. For three of these cases, we visualize the isosurfaces for the FNO prediction and its OpenFOAM ground truth counterpart, with the former color-coded by the distance between each vertex on the FNO isosurface and its closest neighbor on the OpenFOAM isosurface (i.e., representing an error "heat map").

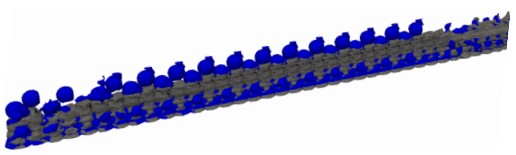

Figure 5: Prediction of an arrangement of stacked droplet lines with $S_{\mathrm{norm}} = 89.61\%$ by FNO models trained on mixed pyramid/hollow cylinder data (dark gray) and pure pyramid data (blue). Adding the hollow cylinder data improves FNO's learning of steep-wall scenarios, a crucial step in enabling it to better predict thin features.

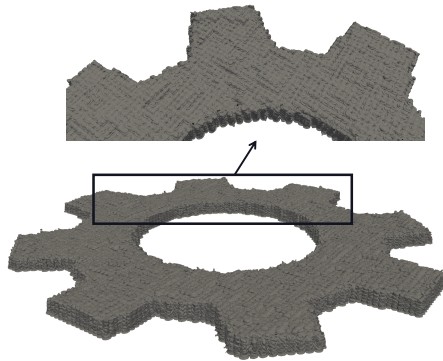

Figure 6: FNO prediction of a gear shape consisting of 16,000 droplets deposited with $S_{\mathrm{norm}} = 89.61\%$. The inset shows a more detailed top-down view of the upper section.

stacked droplet lines, showing promising results for part-scale simulations via a moving subdomain approach.

Our analysis yielded the following major conclusions:

1. FNO shows signs of sufficient out-of-training predictive capability for LMJ. Diversifying the training set with various geometric features (e.g., both infill and thin-wall artifacts) can improve the predictive capability of FNO for build simulation of complex parts, while reducing the amount of data required for training.

2. FNO can accurately predict lines of sequentially deposited droplets for droplet spacings either smaller or bigger than the droplet diameter.

3. FNO is qualitatively capable of predicting thin-wall features generated by stacked lines of droplets and the resulting simple part shapes.

Future activities may include adding physics-based regularization into the FNO training loss to ensure compatibility with relevant conservation laws, and to check whether

this can further reduce the amount of training data needed to achieve a given prediction error. We also plan to compare with other OL approaches such as DeepONet (Lu, Jin, and Pang 2021) to investigate the impact of the NN architecture on generalizability.

While this study addresses prediction of geometric features pertinent to dimensional accuracy and surface quality of as-printed parts, the extension of the predictions to more complex material properties such as residual stresses, elongation, and tensile/compressive strength remains to be investigated. Such predictions will inevitably require including more physical quantities (e.g., temperature fields) in the input/output sets, necessitating further changes in the NN architecture to incorporate multiple inputs and outputs.

# Appendix : Fourier Neural Operator (FNO) architecture

Here we briefly overview the architecture of FNO. More details can be found in (Li et al. 2021). As illustrated in Fig. 3, the mapping from input $a(\mathbf{x})$ to output $b(\mathbf{x})$ consists of the following steps:

1. Lift the input $a(\mathbf{x})$ to a higher-dimensional space through a fully-connected NN representing the local (pointwise) transformation $v_0 = P(a)$.

2. Apply iteratively

$$v_{t+1}(\mathbf{x}) = \sigma \left( W v_t(\mathbf{x}) + (\mathcal{K}(a;\phi)v_t)(\mathbf{x}) \right), \quad (2)$$

for $\mathbf{x} \in \Omega \subset \mathbb{R}^d$. Here $v_t$ ($t = 0, \ldots, T-1$) is a sequence of functions taking values in $\mathbb{R}^{d_v}$, $W : \mathbb{R}^{d_v} \to \mathbb{R}^{d_v}$ is a linear transformation, and $\sigma : \mathbb{R} \to \mathbb{R}$ is a nonlinear activation function applied component-wise.

3. Project back the result $v_T$ into the original space through a fully-connected NN representing the local transformation $b = Q(v_T)$.

In Eq. (2), $\mathcal{K}$ is a kernel integral operator mapping given by:

$$(\mathcal{K}(a;\phi)v_t)(\mathbf{x}) := \int_D \kappa_\phi(\mathbf{x},\mathbf{y},a(\mathbf{x}),a(\mathbf{y}))v_t(\mathbf{y})d\mathbf{y}, \quad (3)$$

where $\mathbf{x}, \mathbf{y} \in \Omega$. Both $W$ and the parameters $\phi$ in the kernel $\kappa_\phi : \mathbb{R}^{2(d+d_a)} \to \mathbb{R}^{d_v \times d_v}$ are learned from data.

To improve the efficiency of their algorithm, (Li et al. 2021) assumed $\mathcal{K}$ to be a convolution operator which, through the convolution theorem, enabled parametrization of $\kappa_\phi$ directly in the Fourier domain. When the domain $\Omega$ is discretized uniformly, this can be done via FFT, accelerated via GPU parallel computing.

## Acknowledgments

The authors are grateful to Zongyi Li (Caltech) for generously sharing the FNO code and helpful comments.

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
