# OpenReview forum: "Accelerating Part-Scale Simulation in Liquid Metal Jet Additive Manufacturing via Operator Learning"
_AAAI.org/2022/Workshop/ADAM — AAAI 2022 Workshop ADAM_

### Official Review · Reviewer_uvTA · 2021-11-30
**The paper presents a new Fourier Neural Operator (FNO) approach to accelerate manufacturing simulations**

**Rating:** 7
**Confidence:** 5

**Review:**

The paper presents a new Fourier Neural Operator (FNO) approach to accelerate metal jet droplet deposition manufacturing simulations. The proposed approach is an order of magnitude faster than the existing state-of-the-art reduced-order simulation approach. The preliminary results show that the FNO approach works better for large spacing between droplets but might need further training for smaller droplet spacings. Overall the results look promising.

I have some minor comments. A more detailed timing and accuracy results would make the advantages of the FNO approach more clear. The quality of the figures could be improved to use vector graphics. Finally, as a possible extension of the work, it might be interesting to understand the effect of the FNO architecture on the results.

---

### Official Review · Reviewer_xMZ7 · 2021-12-01
**Timely research with exciting results, more details of computational gain would make the work stronger**

**Rating:** 7
**Confidence:** 3

**Review:**

The paper proposes an operator learning approach to learn a mapping between initial and final states of the droplet coalescence process to enable rapid and accurate part-scale build simulation. Authors compare this approach to previous work and show the impressive acceleration and reduction of the required data for learning. This is very timely research that can have considerable implications for quality control in additive manufacturing.

The paper contains preliminary results for deposition settings and reports the computational gain. However, I would like to see a more detailed analysis beyond the demonstration of the two cases. For example, the authors state that only 729 pairs were used for training with the proposed approach. How were these 729 pairs selected is unclear.